# Sequence Transfer Learning for Neural Decoding

## Abstract

A fundamental challenge in designing brain-computer interfaces (BCIs) is decoding behavior from time-varying neural oscillations. In typical applications, decoders are constructed for individual subjects and with limited data leading to restrictions on the types of models that can be utilized. Currently, the best performing decoders are typically linear models capable of utilizing rigid timing constraints with limited training data. Here we demonstrate the use of Long Short-Term Memory (LSTM) networks to take advantage of the temporal information present in sequential neural data collected from subjects implanted with electrocorticographic (ECoG) electrode arrays performing a finger flexion task. Our constructed models are capable of achieving accuracies that are comparable to existing techniques while also being robust to variation in sample data size. Moreover, we utilize the LSTM networks and an affine transformation layer to construct a novel architecture for transfer learning. We demonstrate that in scenarios where only the affine transform is learned for a new subject, it is possible to achieve results comparable to existing state-of-the-art techniques. The notable advantage is the increased stability of the model during training on novel subjects. Relaxing the constraint of only training the affine transformation, we establish our model as capable of exceeding performance of current models across all training data sizes. Overall, this work demonstrates that LSTMs are a versatile model that can accurately capture temporal patterns in neural data and can provide a foundation for transfer learning in neural decoding.

## 1 Introduction

A fundamental goal for brain-computer interfaces (BCIs) is decoding intent. Neural decoders that predict behavior address this goal by using algorithms to classify neural patterns. The performance of a decoder is directly related to its ability to overcome sensor noise and intrinsic variability in neural response, particularly in situations with constraints on training data availability. Due to limitations imposed in clinical recording settings, available human subject datasets are commonly on the order of minutes to tens of minutes.

Limited by dataset duration, existing neural decoders achieve reasonable performance by focusing on constrained model designs (Yanagisawa et al., 2012; Vansteensel et al., 2016). As such, the state-of-the art decoders are models which only need to learn a small set of parameters (Gilja et al., 2012; Collinger et al., 2013; Hotson et al., 2016). A limitation of these models is that they rely heavily on the quality of the training data, informative features, and often have rigid timing constraints which limit the ability to model neural variability (Vansteensel et al., 2016). Furthermore, these specific informative features and timing constraints must be hand-tailored to the associated neural prosthetic task and the corresponding neural activity. Deep learning algorithms, however, have been applied to similar problems with the goal of learning more robust representations with large amounts of training data (Graves et al., 2013; Sutskever et al., 2014). Unfortunately, only few studies apply these techniques to neural decoding (Bashivan et al., 2015; Sussillo et al., 2016); however, they exploit different measurement modalities - EEG and single neuron recordings, respectively.

Traditionally hidden Markov models (HMMs) have been used in neural decoding while accounting for temporal variability (Wang et al., 2011; Wissel et al., 2013). Exploring the limitations of these neural decoders and considering recent advancements in deep learning techniques, we propose a

framework using Long Short-Term Memory (LSTM) networks for neural decoding. LSTMs have demonstrated an ability to integrate information across varying timescales and have been used in sequence modeling problems including speech recognition and machine translation (Graves et al., 2013; Sutskever et al., 2014; Cho et al., 2014). While previous work using LSTMs has modeled time varying signals, there has been little focus on applying them to neural signals. To this end, we establish LSTMs as a promising model for decoding neural signals with classification accuracies comparable to that of existing state-of-the-art models even when using limited training data.

Furthermore, addressing the limitations of existing models to generalize across subjects, we propose a sequence transfer learning framework and demonstrate that it is able to exceed the performance of state-of-the-art models. Examining different transfer learning scenarios, we also demonstrate an ability to learn an affine transform to the transferred LSTM that achieves performance comparable to conventional models. Overall, our findings establish LSTMs and transfer learning as powerful techniques that can be used in neural decoding scenarios that are more data constrained than typical problems tackled using deep learning.

## 2 METHODS

### 2.1 DATA DESCRIPTION

Neural signals were recorded from nine subjects being treated for medically-refractory epilepsy using standard sub-dural clinical electrocorticography (ECoG) grids. The experiment was a finger flexion task where subjects wearing a data glove were asked to flex a finger for two seconds based on a visual cue (Miller et al., 2012). Three subjects are excluded from the analysis due to mismatches in behavioral measurement and cue markers. Rejecting electrodes containing signals that exceed two standard deviations from the mean signal, two additional subjects are removed from analysis due to insufficient coverage of the sensorimotor region. All subjects participated in a purely voluntary manner, after providing informed written consent, under experimental protocols approved by the Institutional Review Board of the University of Washington. All subject data was anonymized according to IRB protocol, in accordance with HIPAA mandate. This data has been released publicly (Miller & Ojemann, 2016).

Analyzing the neural data from the four remaining subjects, electrodes are rejected using the same criteria mentioned above. For each subject, 6 - 8 electrodes covering the sensorimotor region are utilized for their importance in motor planning. They are conditioned to eliminate line noise, and then instantaneous spectral power in the high frequency band range (70 - 150 Hz) is extracted for use as the classifier features (Miller et al., 2007). The data is segmented using only the cue information resulting in 27 - 29 trials per finger (5 classes). The signal power for each trial is binned at 150 ms yielding an average sequence length of 14 samples.

As model accuracy is evaluated as a function of the number of training samples, shuffled data is randomly partitioned into train and test sets according to the evaluated training sample count. For each subject, 25 - 27 training samples per class were used allowing for a test set comprising of at least 2 samples per class. A validation set is not used due to the limited data size. All experiments report the average of 20 random partitions of the shuffled data.

### 2.2 BASELINE MODELS

As linear discriminant analysis (LDA) and Hidden Markov models with Gaussian emissions (HMM) are commonly used to decode sequence neural signals (Hotson et al., 2016; Wang et al., 2011; Wissel et al., 2013), we use them for baseline comparisons. Inputs to the HMMs are identical to those used by the LSTMs. However, as LDA does not have a representation of time, individual time bins for each electrode are presented as features to the model. LDA was regularized by automatic covariance shrinkage using Ledoit-Wolf lemma (Ledoit & Wolf, 2004) without using a separate validation set.

The formulation of LDA using time bins as features has been shown to achieve high accuracies with longer trial lengths (Hotson et al., 2016). As such, we emphasize that this formulation results in a strong baseline performance because it can explicitly model the temporal response as a function of neural activity relative to movement onset. To handle varying sequence lengths, an LDA model must be trained for each of the possible sequence lengths. It would be infeasible to construct a

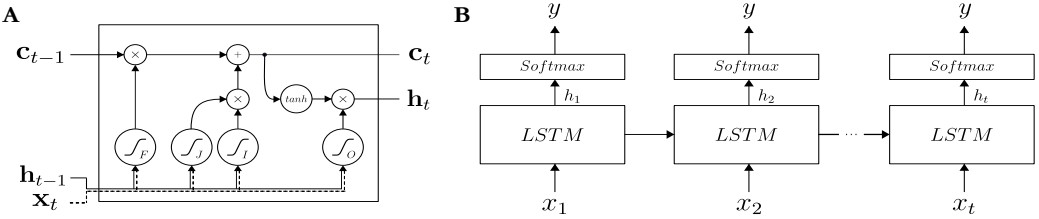

Figure 1: A schematic depicting the Long Short-Term Memory (LSTM) model architecture. (A) Gating and activation functions for a single unit. (B) Unrolled LSTM network with Softmax output at every time step during training.

model for every possible sequence length, and is susceptible to overfitting as electrode count and trial duration increase. This is in contrast to the HMM and LSTM models for which the learned parameters do not change as a function of time relative to movement onset and, instead, temporal dynamics are captured by parameters that are constant across time. Therefore, while LDA works well for structured experimental tasks, it would not generalize to unstructured naturalistic scenarios.

The HMM baseline model is analyzed with 1-state and 2-states using Gaussian emissions to explore if more complex behaviors can be modeled with a more expressive temporal model. For both the 1 and 2-state HMM, a single HMM is trained per class and maximum likelihood decoding is used to identity the target class. While the 2-state HMM is a standard ergodic HMM (Rabiner, 1989) allowing transitions between both the states, a 1-state HMM is a special case, it does not have any state dynamics and makes an independence assumption for samples across time. Thus, the 1-state HMM is specified by

$$
\begin{aligned}
P(\{\mathbf{x}\}\,|y) &= P(\mathbf{x}_1|y)P(\mathbf{x}_2|y)\dots P(\mathbf{x}_t|y) \\
x_t|y &\sim \mathcal{N}(\mu_y, \mathbf{\Sigma}_y)
\end{aligned}
$$

where $\{\mathbf{x}\}$ denotes the sequential data and $y$ denotes the class.

The 1-state HMM assumes the features over time are independent and identically distributed Gaussian. While this independence assumption is simplistic, it is closely related to the LDA model and thus has fewer parameters to estimate. The 2-state HMM on the other hand processes the data as a true time series model similar to LSTM.

## 2.3 NETWORK ARCHITECTURE AND OPTIMIZATION

The single recurrent LSTM cell proposed by Hochreiter & Schmidhuber (1997) is utilized and shown in Figure 1A. The model is completely specified by

$$
\begin{aligned}
f_t &= \sigma(W_{xf}x_t + W_{hf}h_{t-1} + b_f) \\
j_t &= \tanh(W_{xj}x_t + W_{hj}h_{t-1} + b_j) \\
i_t &= \sigma(W_{xi}x_t + W_{hi}h_{t-1} + b_i) \\
o_t &= \sigma(W_{xo}x_t + W_{ho}h_{t-1} + b_o) \\
c_t &= c_{t-1} \odot f_t + i_t \odot j_t \\
h_t &= \tanh(c_t) \odot o_t
\end{aligned}
$$

where $\sigma$ is the sigmoid function, the $W$ terms are weight matrices, the $b$ terms are biases, and $\odot$ represents Hadamard multiplication. To enable gradient flow, the forget gate bias term is initialized to 1 (Gers et al., 2000; Jozefowicz et al., 2015). At every time step during training, the label is provided to allow error propagation during intermediate steps rather than only after the full sequence has been evaluated (Yue-Hei Ng et al., 2015). This is depicted in Figure 1B.

Different weight initialization schemes were analyzed: randomly sampling from a zero-mean Gaussian distribution (Glorot & Bengio, 2010), using a language model (Dai & Le, 2015), and using a sequence autoencoder (Dai & Le, 2015). Building on the sequence autoencoder, we utilize a modified version where the encoder and decoder weights are not shared, and is similar to the LSTM autoencoder from Srivastava et al. (2015). Additionally, dropout is utilized in the recurrent states (Gal

& Ghahramani, 2016) with a probability of 0.3, but is not used on the input layer due to the limited dimensionality. Different model complexities were also explored focusing on network sizes and stacked architectures. The model used for our analyses was constructed with 100 hidden units with no performance gain identified using larger or stacked networks. Gated recurrent units (Cho et al., 2014) were also evaluated with no observed improvement in performance.

Two curriculum learning strategies were identified as potential approaches for improving LSTM trainability. The standard approach uses the full-sequence during training across all epochs. Curriculum learning, originally proposed by Bengio et al. (2009), increases the difficulty of the training samples across epochs and Zaremba & Sutskever (2014) proposed adding a probability for selecting the full-sequence or the curriculum-sequence. This approach is especially useful in situations where optimization using stochastic gradient descent is difficult (Zaremba & Sutskever, 2014). In our task, difficulty is directly related to the length of the sequence. Defining $L$ as the full-sequence length, the curriculum-sequence length, $l$, is scaled by the current epoch, $k$, using $l = \min\{k/8 + 3, L\}$. Each sample in a batch is randomly chosen to be of length $L$ or $l$ with equal probability.

To prevent overfitting to the training set, training was stopped at 75 epochs for all evaluations and subjects. Models were trained through backpropagation using Adam (Kingma & Ba, 2014) and prediction is performed on the test data with accuracies reported using classification at the last time step, except when specified otherwise.

Briefly summarizing the LSTM model: a single single layer of 100 hidden units with 0.3 probability hidden unit dropout and weights initialized by a sequence autoencoder is used. Curriculum learning strategy of Zaremba & Sutskever (2014) with the parameters mentioned was employed for presenting the training samples. All hyperparameters including the number of epochs to stop training were obtained by optimizing a model on 75% of subject Bs data. These selected parameters were kept the same for all evaluations across all subjects. We emphasize that the chosen hyperparameters, though potentially suboptimal, help train a model that generalizes well across all subjects. For reporting accuracies we average results from an evaluation of 20 random partitions of the data. The standard error is at less than 0.02 for all the reported accuracies.

### 2.4 Transfer Learning Architecture

We next explored the possibility of transferring the representation learned by the LSTM from a subject, $S_1$, onto a new subject, $S_2$. Typical transfer learning approaches keep the lower-level layers fixed, but retrain the higher-level layers (Srivastava et al., 2015; Yosinski et al., 2014). Due to the unique electrode coverage and count as well as physiological variability across subjects, this approach yielded poor results. Accounting for these factors, we propose using an affine transform to project the data from $S_2$ onto the input of an LSTM trained on $S_1$ as we might expect a similar *mixture* of underlying neural dynamics across subjects ( Morioka et al. (2015)). The fully connected affine transformation is specified by

$$x_t^{S_2'} \quad = \quad W_{xx} x_t^{S_2} + b_x$$

where $W_x$ and $b_x$ are the weights and biases of the affine mapping, $x_t^{S_2}$ and $x_t^{S_2'}$ are the original and the transformed sequential data from $S_2$ respectively.

Using hyper-parameters outlined in the single subject LSTM model, a two-step training process, shown in Figure 2A, is utilized. The first step trains the LSTM using all of the data from $S_1$. Upon fixing the learned weights, the fully connected affine layer is attached to the inputs of the LSTM and trained on $S_2$ data. To establish that the affine transform is only learning an input mapping and not representing the neural dynamics, a baseline comparison is utilized where the Step 1 LSTM is fixed to a random initialization and only the $Softmax^{S_1}$ is trained, this is shown in Figure 2B.

*All code will be made available.*

## 3 Results

We found that LSTMs trained on an individual subject perform comparable to state-of-the-art models with sufficient training samples. Additionally, using the proposed transfer learning framework we

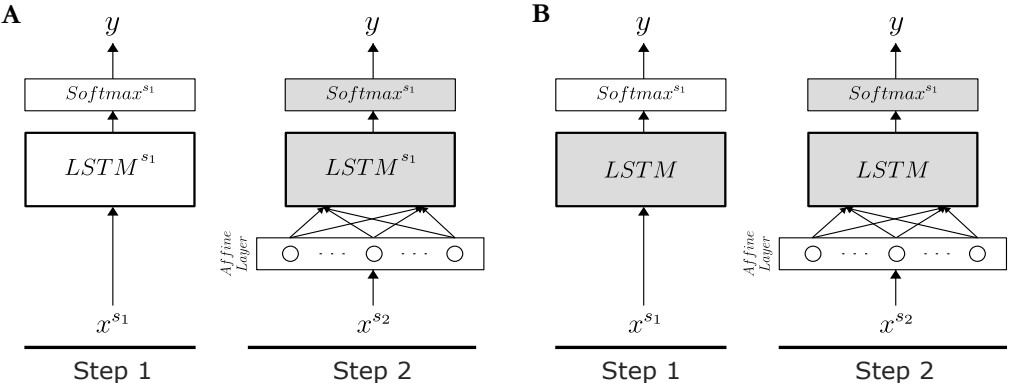

Figure 2: A schematic depicting the LSTM transfer learning model architecture. Where gray indicates fixed and white indicates learned weights, both (A) and (B) depict a 2-step training process with Subject 1 ($S_1$) training in step 1 and Subject 2 ($S_2$) training of only the affine layer in step 2. (A) Transfer learning model (TL) training the LSTM and Softmax layers for $S_1$ in step 1. (B) Randomly initialized LSTM layer and training only the Softmax layer for $S_1$ in step 1.

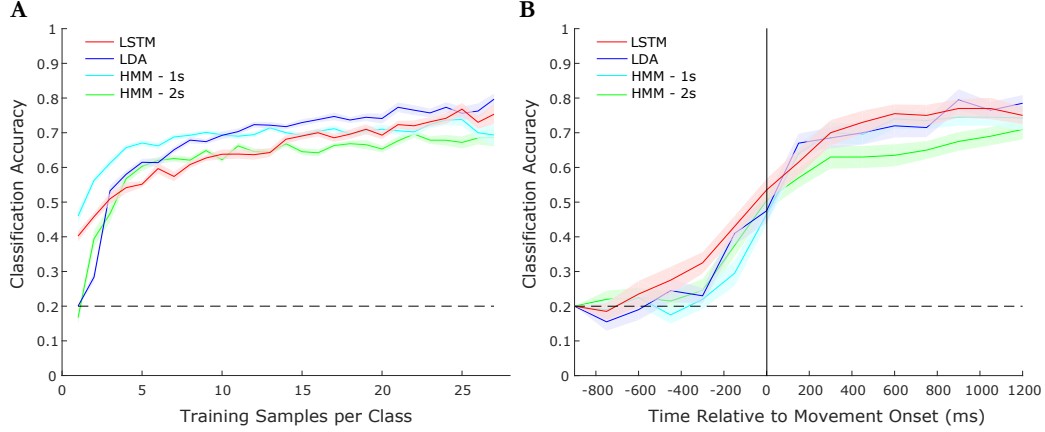

Figure 3: Model performance comparison for representative Subject C. (A) Accuracy as a function of the amount of training samples and (B) Accuracy as a function of time with respect to movement onset evaluated for different models and using all available training data. Error bars show standard error of mean using 20 random partitions of the shuffled data.

observe that LSTMs provide a principled, robust, and scalable approach for decoding neural signals that can exceed performance of state-of-the-art models.

Table 1: Summary comparing the average accuracy of the LSTM model with existing approaches after 20 random partitions of all the shuffled training data.

| Model | Subject A | Subject B | Subject C | Subject D |
|---|---|---|---|---|
| LDA | 0.50 | 0.53 | **0.79** | 0.64 |
| HMM - 1s | 0.51 | 0.61 | 0.69 | 0.65 |
| HMM - 2s | **0.53** | 0.59 | 0.68 | 0.60 |
| LSTM | 0.51 | **0.62** | 0.75 | **0.69** |

### 3.1 LSTM PERFORMANCE

First we establish the performance of the baseline models (Table 1). Interestingly, we observe that increasing the complexity of the HMM marks little improvement in the classification accuracy and typically results in decreased accuracy at low sample counts due to the increased complexity. Additionally, while LDA performs comparably for three subjects, it performs much better than the other models for Subject C due to the increased alignment between cue and observed behavior. This is expected because, as noted, the LDA formulation is better suited to take advantage of the time alignment in the experiment.

Examining the performance of the LSTMs (Table 1), we demonstrate that the proposed model is better able to extract information from the temporal variability in the signals than HMMs and is able to achieve performance comparable to the best baseline for each of the subjects. Consequently, we observe across most subjects LSTMs are able to exceed performance of both HMM models and LDA. Even for Subject C, the LSTM model is comparable to LDA without making the temporal alignment assumption.

Further investigating model time dependence, accuracy is evaluated using neural activity preceding and following behavioral onset. As LDA explicitly models time, a series of models for each possible sequence length are constructed. Depicted in Figure 3B, we observe that the LSTM is slightly better able to predict the behavioral class at earlier times compared to HMMs and is comparable to LDA across all times.

### 3.2 TRANSFER LEARNING PERFORMANCE

Historically, neural prostheses must be tuned frequently for individual subjects to account for neural variability (Simeral et al., 2011; Pandarinath et al., 2017). Establishing LSTMs as a suitable model for decoding neural signals, we explored their ability to learn more robust, generalized representations that can be utilized across subjects.

We demonstrate that learning the affine transformation for the input, it is possible to relax the constraint of knowing exactly where the electrodes are located without having to retrain the entire network. First examining the baseline condition in order to assess the ability for the affine layer to learn the underlying neural dynamics, the $S_1$ LSTM weights were randomly initialized and fixed as outlined in Figure 2B. Looking at Figure 4A, the model performs slightly above chance, but clearly is unable to predict behavior from the data. The TL model where only the affine layer is trained in Step 2 for $S_2$ (Figure 2A) performs comparably to the subject specific model for $S_2$. The notable advantage provided by TL is that there is an increase in loss stability over epochs, which indicates a robustness to overfitting. Finally, relaxing the fixed $LSTM^{S_1}$ and $Softmax^{S_1}$ constraints, we demonstrate that the TL-Finetuned model achieves significantly better accuracy than both the best existing model and subject specific LSTM.

Detailing the performance for each TL and TL-Finetuned, we evaluate all 3 remaining subjects for each $S_2$. For the TL model, we found that the transfer between subjects is agnostic of $S_1$ specific training and performs similarly across all 3 subjects. The performance of TL-Finetuned is similarly evaluated, but has trends unique to TL. In particular, we observe that transferring from Subject A always provides the best results followed by transferring from Subject B. Accuracies using the maximum permissible training data for all four subjects comparing the two transfer learning approaches and the best existing model as well as the subject specific LSTM are reported in Table 2.

### 3.3 MODEL ANALYSIS

For the transfer learning models, we explored the stability of the affine layer and analyzed the learned LSTM weights between subjects. Examining the learned affine mapping, we can see that the layer resembles a projection of electrodes to the appropriate brain regions. In Figure 5A, we show the mapping for two cases: a projection from sensorimotor cortex ($R_{Motor}$, $R_{Sensory}$) between the two subjects, and a projection also adding the occipital lobe electrodes ($R_{Occipital}$) for $S_2$. As the occipital lobe is involved with integrating visual information, and contains information unique from the trained regions, we would expect and do observe, there to be an affine transformation with weights for $R_{Occipital}$ closer to zero indicating an absence of electrodes in $S_1$ that contain similar information. Considering the findings by Haufe et al. (2014), it is important to note that

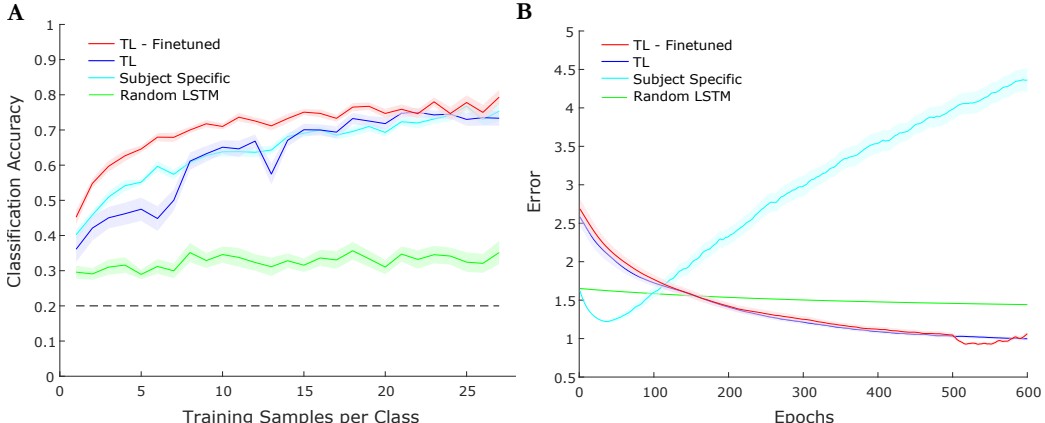

Figure 4: Transfer learning LSTM performance comparison for a representative subject. The TL-Finetuned, TL, and random LSTM all utilize Subject B as $S_1$ and Subject C as $S_2$. The subject specific model uses Subject C. (A) Accuracy as a function of the amount of training samples. (B) Cross-entropy error on the test set across epochs, using 10 training samples per class. Error bars show standard error of mean averaging results from 20 random partitions of the shuffled data.

Table 2: Summary comparing the average accuracy of transfer learning with subject specific training, using all training data across 20 random partitions of the shuffled data.

| Model | Subject A | Subject B | Subject C | Subject D |
|---|---|---|---|---|
| Best Existing Model | 0.53 | 0.61 | 0.79 | 0.65 |
| Subject Specific | 0.51 | 0.62 | 0.75 | 0.69 |
| TL ($S_1$ = Subject A) | - | 0.66 | 0.72 | 0.60 |
| TL ($S_1$ = Subject B) | 0.40 | - | 0.70 | 0.46 |
| TL ($S_1$ = Subject C) | 0.44 | 0.63 | - | 0.59 |
| TL ($S_1$ = Subject D) | 0.44 | 0.67 | 0.73 | - |
| TL-Finetuned ($S_1$ = Subject A) | - | **0.71** | **0.82** | **0.70** |
| TL-Finetuned ($S_1$ = Subject B) | 0.46 | - | 0.75 | 0.62 |
| TL-Finetuned ($S_1$ = Subject C) | 0.44 | 0.63 | - | 0.66 |
| TL-Finetuned ($S_1$ = Subject D) | **0.53** | **0.71** | 0.79 | - |

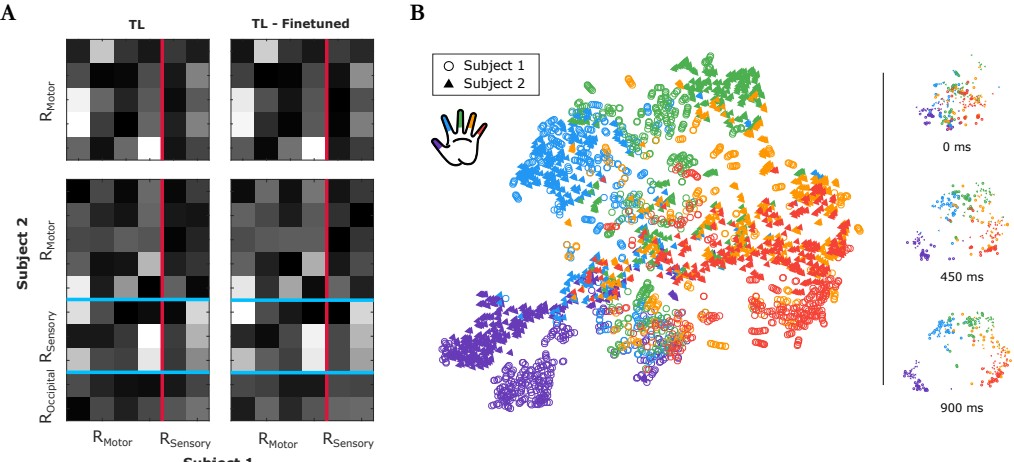

Figure 5: Visualizations of model parameters. (A) Weights of the learned affine mapping from $S_2$ (Subject C) to $S_1$ (Subject B). Electrodes mapped from motor to sensorimotor and sensorimotor + occipital to sensorimotor regions of $S_2$ to $S_1$, respectively. Both TL and TL-Finetuned weights are depicted. (B) Two-dimensional t-SNE embedding of the learned signal representation corresponding to different fingers for both subjects.

we know apriori from neurophysiology that the occipital region in motor movement tasks should be less informative. Therefore, we would expect the affine mapping to learn to reject the signals from occipital region which we observe in our model. Additionally, we can qualitatively observe the stability of the affine transform after the fixed $LSTM^{S_1}$ and $Softmax^{S_1}$ are relaxed. It is clear by looking between the left and right columns of Figure 5A that the learned weights from TL are a good representation and only require minor modification in TL-Finetuned.

Furthermore, exploring the use of LSTMs for transfer learning, a two-dimensional embedding of the LSTM output using t-SNE (Maaten & Hinton, 2008) on the training data was created. We use the $LSTM^{S_1}$ outputs for $S_1$ data and $LSTM^{S_1}$ outputs for $S_2$ data after passing it through the learned affine layer on $S_2$. All the data between -300 ms to 900 ms, relative to behavioral onset, is embedded in the two-dimensional space. From the summary image, Figure 5B, it is clear that the model is able to separate classes for both subjects well and that the projections for both the subjects are clustered together. Identifying the source of the noise in the figure, we project the output at different time steps and see that the majority of the confusion occurs at the start of the sequences.

## 4 DISCUSSION

Although many neural decoding models and processing techniques have been proposed, little work has focused on efficient modeling of time-varying dynamics of neural signals. In this work, we have shown that LSTMs can model the variation within a neural sequence and are a good alternative to state-of-the-art decoders. Even with a low sample count and comparably greater number of parameters, the model is able to extract useful information without overfitting. Moreover, LSTMs provide a robust framework that is capable of scaling with large sample counts as opposed to the the limited scalability provided by existing approaches. Considering the limitations imposed on our model by stopping at a fixed evaluation epoch, it would be possible to further boost performance by utilizing early stopping with a validation set. And while the input features were selected from empirical observations made in previous studies, the results could be improved by extracting the features in an unsupervised manner using autoencoders (Poultney et al., 2007; Le et al., 2011) or by training the decoder end-to-end using convolutional LSTMs (Shi et al., 2015; Zhang et al., 2016).

Establishing the LSTM as a good approach for neural decoding, we explored utilizing the model in transfer learning scenarios. Exploring a less constrained model where the LSTM weights are relaxed, the performance is shown to exceed that of both the subject specific training and the best

decoder models. This robustness against subject specific neural dynamics even when only the affine transform is learned indicates that the LSTM is capable of extracting useful information that generalizes to the new subject with limited impact due to the original subject's relative performance. When exploring the tradeoffs between TL and TL-Finetuned, the latter provides performance that exceeds the current state-of-the-art models with fewer subject-specific training samples where the former is able to achieve comparable performance. However, because TL only requires training of the affine layer, the computation is less expensive than the TL-Finetuned. From Figure 4, it could be seen when learning only the affine transformation the cross-entropy loss is still decreasing after 500 epochs suggesting that with better optimization methods, the TL model by itself may outperform the subject-specific model. This motivates the statement that the LSTM is capable of extracting a representation of the neural variability between behaviors that generalizes across subjects. While this may be specific to the behavior being measured, it posits potential scenarios for using sequence transfer learning.

Exploring the reasoning behind the affine layer, we consider relaxing the structured mapping of electrodes between the subjects required by typical transfer learning models. While the structured mapping would intuitively yield good results if electrode placement is the sole factor influencing neural variability, we see that it leads to suboptimal performance due to limited alignment capabilities as well as the underlying neural representation being unique. The addition of an affine layer, however, provides sufficient flexibility for the input remapping to account for this variability and matches the new subjects input electrodes based on error minimization. Moreover, the weights that are produced through backpropagation are able to optimize the weights regardless of $S_2$ input dimensionality and thus allows for eliminating use of non-informative electrodes. This input gating leads to model training stability and is shown to be a valid mapping due to the minimal weight update when transitioning from TL to TL-Finetuned. Furthermore, exploring the relationship between subjects 1 and 2, the t-SNE analysis shows that the learned parameters for the affine layer provide a meaningful mapping between the two subjects likely indicating an underlying physiological basis rooted in the structure of the sensorimotor cortex that has been shown to exist.

We believe that the approaches established in this paper provide techniques for decoding neural signals that had not yet been explored in detail. Particularly, the insights gained from exploring neural decoding leveraging the expressibility and generalizability of LSTMs yielded techniques that provide more accurate and robust models compared to current state-of-the-art decoders. Consequently, the strategies of applying LSTMs to sequence learning and sequence transfer learning problems will be useful in a variety of neural systems.

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
