# OpenReview forum: "Sequence Transfer Learning for Neural Decoding"
_ICLR.cc/2018/Conference — Reject_

### Official Review · AnonReviewer3 · 2017-11-25
**Difficult problem, some aspects are unclear, evaluation could be improved**

**Rating:** 4
**Confidence:** 5

**Review:**

The paper describes an approach to use LSTM’s for finger classification based on ECOG. and a transfer learning extension of which two variations exists. From the presented results, the LSTM model is not an improvement over a basic linear model. The transfer learning models performs better than subject specific models on a subset of the subjects. Overall, I think the problem Is interesting but the technical description and the evaluation can be improved. I am not confident in the analysis of the model. Additionally, the citations are not always correct and some related work is not referenced at all. For the reasons above, I am not willing to recommend the paper for acceptance at his point.

The paper tackles a problem that is challenging and interesting. Unfortunately, the dataset size is limited.
This is common for brain data and makes evaluation much more difficult.
 The paper states that all hyper-parameters were optimized on 75% of subject B data.
The actual model training was done using cross-validation.
So far this approach seems more or less correct but in this case I would argue that subject B should not be considered for evaluation since its data is heavily used for hyper-parameter optimization and the results obtained on this subject are at risk of being biased.
Omitting subject B from the analysis, each non-transfer learning method  performs best on one of the remaining subjects.
Therefore it is not clear that an LSTM model is an improvement.
For transfer learning (ignoring B again) only C and D are improved but it is unclear what the variance is.
In the BCI community there are many approaches that use transfer learning with linear models. I think that it would be interesting how linear model transfer learning would fare in this task.

A second issue that might inflate the results is the fact that the data is shuffled randomly. While this is common practice for most machine learning tasks, it is dangerous when working with brain data due to changes in the signal over time. As a result, selecting random samples might inflate the accuracy compared to having a proper train and test set that are separated in time. Ideally the cross-validation should be done using contiguous folds.

I am not quite sure whether it should be possible to have an accuracy above chance level half a second before movement onset? How long does motor preparation take? I am not familiar with this specific subject, but a quick search gave me a reaction time for sprinters of .15 seconds. Is it possible that cue processing activity was used to obtain the classification result? Please discuss this effect because I am do not understand why it should be possible to get above chance level accuracy half a second before movement onset.

There are also several technical aspects that are not clear to me. I am confident that I am unable to re-implement the proposed method and their baseline given the information provided.

LDA baseline:
—————————
For the LDA baseline, how is the varying sequence length treated?
Ledoit wolf analytic  regularization is used, but it isn not referenced. If you use that method, cite the paper.
The claim that LDA works for structured experimental tasks but not in naturalistic scenarios and will not generalize when electrode count and trial duration increases is a statement that might be true. However, it is never empirically verified.  Therefore this statement should not be in the paper.

HMM baseline
—————————
How are the 1 and the 2 state HMM used w.r.t. the 5 classes? It is unclear to me how they are used exactly. Is there a single HMM per class? Please be specific.

LSTM Model
—————
What is the random and language model initialization scheme? I can only find the sequence auto-encoder in the Dai and Le paper.


Model analysis
——————————-
It is widely accepted in the neuroimaging community that linear weight vectors should not be interpreted directly. It is actually impossible to do this.  Therefore this section should be completely re-done. Please read the following paper on this subject.
Haufe, Stefan, et al. "On the interpretation of weight vectors of linear models in multivariate neuroimaging." Neuroimage 87 (2014): 96-110.

References
————
Ledoit wolf regularization is used but not cited. Fix this.
There is no citation for the random/language model initialization of the LSTM model. I have no clue how to do this without proper citation.
Le at al (2011) are referenced for auto-encoders. This is definitely not the right citation.
Rumelhart, Hinton, & Williams, 1986a; Bourlard & Kamp, 1988; Hinton & Zemel, 1994 and Bengio, Lamblin, Popovici, & Larochelle, 2007; Ranzato, Poultney, Chopra, & LeCun, 2007 are probably all more relevant.
Please cite the relevant work on affine transformations for transfer learning especially the work by morioka et al who also learn an input transferm.
Morioka, Hiroshi, et al. "Learning a common dictionary for subject-transfer decoding with resting calibration." NeuroImage 111 (2015): 167-178.

---

> ### Author Response · Authors · 2017-12-23
> **Response to Reviewer3 continued**
>
> > “For the LDA baseline, how is the varying sequence length treated?”; “The claim that LDA works for structured experimental tasks but not in naturalistic scenarios and will not generalize when electrode count and trial duration increases is a statement that might be true. However, it is never empirically verified. Therefore this statement should not be in the paper.”
> For each of the possible sequence lengths an LDA model is trained, thus, a family of LDA models is constructed and the model that is picked depends on the length of the sequence. This is the reasoning behind the claim that LDA would not generalize when trial duration increases, as it would be infeasible to construct an LDA model for every possible sequence length.
>
> > “How are the 1 and the 2 state HMM used w.r.t. the 5 classes?”
> There is a single HMM per class; we have updated the manuscript to make this clear.
>
> > “What is the random and language model initialization scheme?”
> With respect to the LSTM initialization schemes: by random we mean initializing weights of the network randomly, say using Xavier initialization (Glorot and Bengio 2010); language model initialization is from the Dai and Le paper. We have updated to make this clear in the manuscript.
>
> > Model analysis
> Thank you for the insightful reference; reviewing the statements of Haufe et al and our results, we think our interpretations are still valid, but need to be reworded in the manuscript to prevent misinterpretation. While we are utilizing a backward model, we are not hoping to make conclusions about the learned weights in the affine mapping specifically concerning the underlying brain processes. We are addressing the fact that the LSTM model input features are electrodes with specific locations on Subject 1, and when we include known non-informative electrodes, the learned representation excludes them. I.e. we know from the task design and neurophysiology that the occipital region in motor movement tasks should have little activation. Therefore, we would expect the affine mapping to learn zero weights for the signals from occipital region and we do see that in our model. We believe that this interpretation does not violate the points from Haufe et al.
>
> > References
> Our apologies for the missing references. We have included the citations for the Ledoit-Wolf lemma and the prior transfer learning work in EEG (Morioka et al.). In the discussion section we refer to usage of auto-encoders as an unsupervised feature extractor, this is in contrast to our current features which are based on neurophysiology, and hence the reference to Le et al 2011.

---

> > ### Comment · AnonReviewer3 · 2018-01-09
> > **Thank you for the clarifications**
> >
> > I will response to all relevant comments here.
> >
> > w.r.t. Subject B
> > - Here I did not change my opinion.
> > The hyper-parameters are optimized on the data that is used for evaluation. This is a basic machine learning error. Therefore the best thing would be to exclude the data from subject B since there is some doubt about the validity of those results.
> >
> > w.r.t. the LDA model
> > - Thanks for the clarification. This approach is more complex than I expected.
> >  A different solution to having to train different models of different length could be to view it as a convolutional approach with max or mean pooling to obtain the actual output. Given that it is very easy to implement this, I would encourage the authors to try this.
> > This approach would also have the advantage that more data is available to train the entire model.
> > Also, this would enable the authors to test the affine transformation in combination with the linear model.
> > If the LSTM still performs better, this would make the paper a lot stronger.
> >
> > w.r.t. model analysis
> > - Without a colour map next to the weights I cannot understand the plotted matrix.
> > - The conclusion that zero weights do not contribute to the output is only valid if they are actually zero and not really small. The whole point of the Haufe paper is that a small weight might be important and processing information, while a large weight might be there primarily due to noise cancelling.
> > - Since there is no information about whether the weights or zero or just small, I cannot conclude anything from the provided data.
> > - That being said, I am not convinced that the paper needs Fig 5A. Making fig 5b larger would be more informative.
> >
> > w.r.t. language model init.
> > It is still not clear to me how exactly the model is pre-trained. The notion of language model does not exist here and this might be confusing me. Is it just an auto-encoder trained to predict the data one time-step ahead?
> >
> > w.r.t. relation to other TL approaches.
> > I understand that the difference between EEG and ECOG is that ECOG is typically placed on different parts of the brain. What I fail to grasp is why an affine transformation makes sense here. Is there an intuition about how this can compensate for electrode placement?
> >
> > Finally, another suggestion.
> > If transfer learning works well, it would make sense to jointly train on all subjects. Where each subject has its own affine transformation but where the model after the transformation is shared across subjects.

---

> ### Author Response · Authors · 2017-12-23
> **Response to Reviewer3**
>
> Thank you for your detailed remarks - we hope to address all of your concerns below and in the updated manuscript.
>
> > “I would argue that subject B should not be considered for evaluation since its data is heavily used for hyper-parameter optimization and the results obtained on this subject are at risk of being biased.”
> In an attempt to limit the amount of subject-specific tuning performed for the LSTMs, we limit the hyper-parameter tuning to only Subject B. Regarding the inclusion of Subject B, we limit the amount of fine tuning that is done for on the model to limit the potential bias when evaluating Subject B. Furthermore, we demonstrate that other subjects, using Subject B’s hyperparameters, perform comparably. While tuning hyperparameters will likely improve performance for each subject, that would require setting aside data for hyperparameter tuning in the already data constrained situation. Clarifying the transfer learning results, the variance can be reported with standard error of mean (SE) values where, for all reported accuracies, the SE is at most 0.02.
>
> > “it is not clear that an LSTM model is an improvement”
> As we state in the manuscript, we demonstrate the LSTM model achieves performance comparable to the other models. The primary advantage to utilizing LSTMs resides in the ability to learn the mapping for an affine transformation (please refer to our response to Reviewer2).
>
> > “In the BCI community there are many approaches that use transfer learning with linear models. I think that it would be interesting how linear model transfer learning would fare in this task.”
> In exploring existing techniques for transfer learning in neural data, there are limited approaches that can be applied to ECoG data and none found in existing literature. Typical techniques utilize EEG data which has a significant amount of spatial averaging allowing for more direct mapping between subjects. In ECoG, the array placements are unique for each subject have greater spatial resolution that exploits underlying neural structures (i.e. using only sensorimotor cortex electrodes), but makes it increasingly difficult to directly map between subjects. Regarding the models, specifically, we may be able to adapt them for transfer learning, however, they would be limited by either the need for a hand-tuned affine transform, or would not represent time series.Specifically, exploring the transfer learning capabilities of the other proposed models, it is important to consider the key advantage of LSTMs is that backpropagation allows for learning the affine transform. Modification of the other models would require hand tuning a mapping layer, or constructing an ensemble of models to leverage a learned mapping to make TL possible. While it is possible to extend LDA, a key goal was to move away from models that operate only on fixed time contexts such as LDA, to a time-series model.
>
> > “issue that might inflate the results is the fact that the data is shuffled randomly”
> Regarding the random shuffle, the experimental setup does not require analysis of contiguous data folds. While neural activity changes over time, the duration of the experiment is on the order of tens of minutes, not hours, which should limit the amount of variability present. Furthermore, between each trial, there is a refractory period that allows the baseline dynamics to be achieved.
>
> > “Accuracy above chance level half a second before movement onset”
> This is possible because cue processing activity was used to obtain the classification result. As mentioned in the manuscript, the trials are segmented based on the cue. Hence, accuracy above chance is achieved by integrating over 300 ms of neural activity beginning from the display of the cue. Hotson et al. 2016 also show that it is possible to decode prior to movement onset.

---

### Official Review · AnonReviewer1 · 2017-11-27
**LSTMs for ECoG**

**Rating:** 6
**Confidence:** 5

**Review:**

The ms applies an LSTM on ECoG data and studies tranfer between subjects etc.

The data includes only few samples per class. The validation procedure to obtain the model accuray is a bit iffy.
The ms says: The test data contains 'at least 2 samples per class'. Data of the type analysed is highly dependend, so it is not unclear whether this validation procedure will not provide overoptimistic results. Currently, I do not see evidence for a stable training procedure in the ms. I would be curious also to see a comparison to a k-NN classifier using embedded data to gauge the problem difficulty.
Also, the paper does not really decide whether it is a neuroscience contribution or an ML one. If it were a neuroscience contribution, then it would be important to analyse and understand the LSTM representation and to put it into a biological context fig 5B is a first step in this direction.
If it where a ML contribution, then there should be a comprehensive analysis that indeed the proposed architecture using the 2 steps is actually doing the right thing, i.e. that the method converges to the truth if more and more data is available.
There is also some initial experiments in fig 3A. Currently, I find the paper somewhat unsatisfactory and thus preliminary.

---

> ### Author Response · Authors · 2017-12-23
> **Response to Reviewer1**
>
> Thank you for your remarks - we hope to address all of your concerns below and in the updated manuscript.
>
> > “The data includes only few samples per class. The validation procedure to obtain the model accuray is a bit iffy. Currently, I do not see evidence for a stable training procedure in the ms.”
> The availability of invasive neural recordings in humans is quite limited. Therefore, we selected a validation procedure that demonstrates the robustness of the model training across multiple runs and partitions of data. We show in a the various groupings, a consistent performance is achieved. Furthermore, we did not finetune the parameters for all subjects, rather, a single set of hyperparameters for the model was selected after being coarsely trained on a single subject.
>
> > “Comparison to a k-NN classifier using embedded data to gauge the problem difficulty”
> Regarding the comparison to a k-NN classifier (presumably on the 2-d embedding), the t-SNE embedding is obtained on the training data after the network was optimized on it. As such, the embedding looks separable because it was optimized on this data. Because t-SNE is nonparametric, the test data cannot be projected onto the same embedding space to facilitate the k-NN experiment. We use the t-SNE embedding to show that the learned parameters of the network separate classes for both subjects well and the embedding for the same class from the two subjects cluster together even though the network weights were not explicitly optimized for it.
>
> > “Also, the paper does not really decide whether it is a neuroscience contribution or an ML one.”
> The primary contributions we hope to make clear through the paper are not restricted to neuroscience or machine learning due to the nature of the experimentation and analysis. We hope to demonstrate the benefits and disadvantages of existing approaches, applications of new models that have not been applied to neural data, and to propose improvements to the model demonstrating benefits in performance with potential for understanding what is being learned rooted in neuroscience principles.

---

### Official Review · AnonReviewer2 · 2017-11-27
**Application of LSTM to decoding of neural signals, limited novelty, inconclusive**

**Rating:** 3
**Confidence:** 4

**Review:**

This work addresses brain state  decoding (intent to move) based on intra-cranial "electrocorticography (ECoG) grids". ECoG signals are generally of much higher quality than more conventional EEG signals acquired on the skalp, hence it appears meaningful to invest significant effort to decode.
Preprocessing is only descibed in a few lines in Section 2.1, and the the feature space is unclear (number of variables etc)

Linear discriminants, "1-state and 2-state" hidden markov models, and LSTMs are considered for classification (5 classes, unclear if prior odds are uniform). Data involves multiple subjects (4 selected from a larger pool). Total amount of data unclear. "A validation set is not used due to the limited data size."  The LSTM setup and training follows conventional wisdom.
"The model used for our analyses was constructed with 100 hidden units with no performance gain identified using larger or stacked networks."
A simplistic but interesting  transfer scheme is proposed amounting to an affine transform of features(??) - the complexity of this transform is unclear.

While limited novelty is found in the methodology/engineering - novelty being mainly related to the affine transfer mechanism, results are disappointing.
The decoding performance of the LSTMs does not convincingly exceed that of the simple baselines.

When analyzing the transfer mechanism only the LSTMs are investigated and it remains unclear how well trans works.

There is an interesting visualization (t-SNE) of the latent representations. But very limited discussion of what we learn from it, or how such visualization could  be used to provide neuroscience insights.

In the discussion we find the claim: "In this work, we have shown that LSTMs can model the variation within a neural sequence and are a good alternative to state-of-the-art decoders."  I fail to see how it can be attractive to obtain similar performance with a model of 100x (?) the complexity

---

> ### Author Response · Authors · 2017-12-23
> **Response to Reviewer2**
>
> Thank you for your remarks - we hope to answer all of your concerns below or in the updated manuscript.
>
> > “the feature space is unclear (number of variables etc)”
> Briefly clarifying the feature space, we use high frequency band (70 - 150 Hz) power extracted from the electrodes in the sensorimotor region. The number of features (equivalent to the number of electrodes as a single average power is utilized) vary based on the subject but are between 6 and 8.
>
> > “Unclear if prior odds are uniform. Total amount of data unclear.”
> For the models, the classes are balanced and the number of samples per classes vary based on the subject, but are between 27 and 29. Thus the total number of samples are between 135 and 145 and a uniform prior is utilized.
>
> > “A simplistic but interesting transfer scheme is proposed amounting to an affine transform of features(??) - the complexity of this transform is unclear.”
> Regarding the affine transform, it transforms the feature space from the new target subject to the feature space of the original subject. As the transform is affine, the number of parameters of the transform is e_2*(e_1+1) where e_1, e_2 are the number of electrodes in the original subject and the transferring subject, respectively.
>
> > “The decoding performance of the LSTMs does not convincingly exceed that of the simple baselines.”; “I fail to see how it can be attractive to obtain similar performance with a model of 100x (?) the complexity”
> With respect to the decoding performance of the LSTM’s, we establish the model as an approach that provides comparable results even while having significantly more parameters to learn and being data constrained. An advantage to utilizing such a technique is due to the scalability of the model (as seen in speech recognition literature Graves et al. 2013). The key advantage to the LSTM model, however, is the demonstrated superior performance utilizing the TL architecture which is only possible due to the network structure and the ability to back propagate errors. Furthermore, preventing training of the LSTM and only training the affine mapping provides comparable performance to existing techniques that could have applications where subject specific training data is limited.
>
> > “When analyzing the transfer mechanism only the LSTMs are investigated and it remains unclear how well trans works.”
> Exploring the transfer learning capabilities of the other proposed models, it is important to consider the key advantage of LSTMs is that backpropagation allows for learning the affine transform. Modification of the other models would require hand tuning a mapping layer, or constructing an ensemble of models to leverage a learned mapping to make TL possible. While it is possible to extend LDA, a key goal was to move away from models that operate only on fixed time contexts such as LDA, to a time-series model. We discuss reasoning for this in the manuscript.
>
>  > “There is an interesting visualization (t-SNE) of the latent representations. But very limited discussion of what we learn from it, or how such visualization could be used to provide neuroscience insights.”
> The t-SNE representation is meant to demonstrate that a meaningful mapping between the two subjects exists rather than some arbitrary mapping that gives good results (i.e. the results are interpretable). It is likely a representation of an underlying physiological basis rooted in the structure of the sensorimotor cortex. However, a more detailed examination with more subjects is necessary to be able to concretely say anything about the underlying physiology.

---

### Decision · Program_Chairs · 2018-01-29
**ICLR 2018 Conference Acceptance Decision**

**Decision:**

Reject

**Comment:**

This paper tries to establish that LSTMs are suitable for modeling neural signals from the brain.  However, the committee and most reviewers find that results are inconclusive.  Results are mixed across subjects.  We think it would have been far more interesting to compare other types of sequence models for this task other than the few simple baselines implemented here.  It is also unclear what is the LSTM learning extra in contrast with the other models presented in the paper.